# Bar Load-Velocity Profile of Full Squat and Bench Press Exercises in Young Recreational Athletes

**DOI:** 10.3390/ijerph19116756

**Published:** 2022-06-01

**Authors:** Jairo Alejandro Fernandez Ortega, Dario Mendoza Romero, Hugo Sarmento, Laura Prieto Mondragón

**Affiliations:** 1Faculty of Health Sciences, University of Applied and Environmental Sciences, Street 222 #55-37, Bogotá 111166, Colombia; laprieto@udca.edu.co; 2Laboratory of Exercise Physiology, Faculty of Physical Education, National Pedagogical University, Street 7 #11-86, Bogotá 480100, Colombia; 3Universidad Santo Tomas, North Highway 205 Street, Via Arrayanes km 1.6, Bogotá 110141, Colombia; dariomendoza@usantotomas.edu.co; 4University of Coimbra, Research Unit for Sport and Physical Activity (CIDAF), Faculty of Sport Sciences and Physical Education, 3004-531 Coimbra, Portugal; hg.sarmento@gmail.com

**Keywords:** muscle strength, stretch-shortening cycle, weightlifting, young adult

## Abstract

The purpose of this study was to determine the mean propulsive velocity (MVP) at various percentages of one repetition maximum (1RM) in the full squat and chest press exercises. A total of 96 young women and 256 young men (recreational athletes) performed an incremental test (50–60–70–80% 1RM) comprising the bench press and full squat exercises in two different sessions. The individual load and velocity ratios were established through the MPV. Data were analyzed using SPSS software version 25.0, with the significance level set at 5%. The following findings were revealed: highly linear load-velocity relationships in the group of women (r = 0.806 in the squat, and r = 0.872 in the bench press) and in the group of men (r = 0.832 and r = 0.880, respectively); significant differences (*p* < 0.001) in the MPV at 50–70–80% 1RM between the bench press and the full squat in men and at 70–80% 1RM in women; and a high variability in the MPV (11.49% to 22.63) in the bench press and full squat (11.58% to 25.15%) was observed in women and men (11.31% to 21.06%, and 9.26% to 24.2%) at the different percentages of 1RM evaluated. These results suggest that the load-velocity ratio in non-strength-trained subjects should be determined individually to more precisely establish the relative load to be used in a full squat and bench press training program.

## 1. Introduction

Historically, strength training programs were defined by a percentage of the maximum strength of each subject (1RM) [1]. The role of velocity in the movements performed during strength training was dismissed until recently, even though the strength–velocity relationship has been demonstrated in vitro [2,3] and in vivo [4,5] for more than eight decades.

Mathematical models have been developed based on the load-velocity relationship to estimate the maximum force (expressed as 1RM) [6]. The general load-velocity relationship was introduced by Gonzalez-Badillo and Sanchez-Medina [7], who used a second-order polynomial regression equation to estimate %1RM during the bench press exercise based on velocity [6,8,9,10,11,12,13,14]. The mathematical models use linear regressions between the load values (X) and AV (average velocity; Y). Based on this information, the slope (SLP), the theoretical AV at 0 kg (AVO) and the theoretical load at 0 m/s^−1^ (LD0) are calculated. The use of these variables is argued because the AV allows a greater representation of the subject’s ability to move the load throughout the entire concentric phase, and this variable, by decreasing linearly with the increase in the load, facilitates the mathematical analysis [6].

Since then, much research has been carried out to determine general equations for the velocity–load relationship so that velocity can be used to determine %1RM or 1RM [7,10,15,16,17,18,19]. 

The findings allowed the establishment of a new modality of strength training based on the velocity of execution, which quickly showed positive effects on physical performance (strength, jumping, running velocity, pedaling power) [1,20,21,22,23,24]. Thus, this modality has become a mechanism for prescribing and controlling training loads.

The velocity at which a movement is performed during strength training is critical to the type of adaptations that will occur, as well as the corresponding neuromuscular demands and fatigue levels [1]. However, the velocity values associated with %1RM in a population are not necessarily transferable to other muscle groups or to another population due to various aspects [10,17].

The same velocity represents different intensities or %1RM values depending on the exercise performed. For example, exercises that involve large muscle groups produce a higher velocity for the same 1RM percentages [10,17].

Gender and age are important variables in this context. There are a few studies involving men and women, in which the velocity associated with each %1RM was greater in men [25], and significant differences were identified for the maximum squat velocity at 40, 50, 60 and 70% 1RM; the bench press maximum power at 50, 60, 70 and 80%; the time to reach the maximum velocity of 1RM in the bench press at 50, 60, 70 and 80%; and the time to reach the 1RM bench press maximum power at 50, 60, 70 and 80% [26].

Additionally, younger athletes tend to develop a higher velocity with each %1RM than their older counterparts [27]. Finally, an individual’s load-velocity profile depends on their training level and experience with strength work [28]. Therefore, it is important to accurately determine individualized load-velocity profiles [29].

This way, the optimal muscular strengthening conditions can be determined, thus bringing strength training conditions closer to the specific conditions required during movements.

To the best of our knowledge, no previous study has determined the load-velocity relationship between bench press and full squat exercises in a group of recreational male and female athletes. Therefore, detailed studies are required to identify the behavior of the load-velocity relationship in an untrained population. Thus, the main objective of this study was to investigate the velocity associated with each %1RM in full squat and bench press exercises among young male and female recreational athletes.

## 2. Materials and Methods

Previous studies [7,17,25,30] have associated different velocity values with different %1RM values using mathematical equations in populations among male performance athletes. Here, we attempted to verify whether these values are similar in a population of male and female recreational athletes. Specifically, the MPV was evaluated at 50, 60, 70 and 80% 1RM for chest press movements and full squats.

### 2.1. Data Collection

This randomized control trial involved 355 young adults: 96 women (20.4 ± 2.5 years old, 158.2 ± 17.3 m, 61.6 ± 8.4 kg) and 259 men (20.9 ± 2.5 years old, 170.5 ± 12.6 cm, 65.8 ± 9.2 kg). The participants were physically active students from academic programs related to sports science or physical education. No subject presented physical limitations, health problems or musculoskeletal injuries, and none had participated in any strength training programs.

Participants were informed of the details, objectives, benefits and risks of the study and signed an informed consent form. The study was designed following the ethical standards stipulated in the Declaration of Helsinki and was approved by the research ethics committee of the University of Applied and Environmental Sciences.

### 2.2. Procedures

Before starting the tests, the subjects performed six sessions to become familiar with the equipment and the techniques used to execute the movements. Loads of 13 kg and 20 kg (for women) and 20 kg and 30 kg (for men) were established in the bench press and squats, respectively. During each session, 3 sets of 15 repetitions were performed for each movement. For the full squat, technique and knee flexion angle (~35–40°) were emphasized [31]. In the eccentric phase, these parameters were verified with a goniometer during the slow and controlled descent. In the concentric phase, they were measured based on the ascent maximal voluntary velocity when the athlete returned to an upright position without lifting their heels off the floor. During the familiarization phase, the subjects were instructed to perform concentric actions at maximum speed, and during the tests, the participants were verbally motivated to move the load at maximum velocity in each repetition.

In the bench press, special emphasis was placed on the controlled descent of the bar until 90° elbow flexion, where the width of the bar grip was the biacromial distance, and in this position, they had to wait (approximately 1.5 s) until the evaluator’s order to perform the movement that had to be executed at the highest possible speed, keeping the back on the bench and until reaching full extension.

The tests were carried out across three sessions separated by 72 h [32]. Participants did not carry out any physical training between sessions. In the first session, body composition was assessed, and physical activity, personal data and medical history questionnaires were administered. In the second session, 1RM values were determined, both for the squat and bench press. In the third session, the MPV values were determined using an incremental loading test for both movements. All tests were performed at the same time of day (2:00–4:00 p.m.) to control for the effects of circadian rhythms on neuromuscular performance [33,34].

A warm-up was carried out before each test. Warm-ups consisted of: 5 min of band jogging at 8 km/h, 5 min of active stretching and joint mobility and three series of eight full squat and bench press repetitions using only the weight of the bar. After three minutes of recovery, the maximum force was estimated [35].

Standard procedures for testing 1RM in youth were followed [36]. All exercises were performed in a Smith machine (Life-fitness, Los Angeles, CA, USA), which allows the bar to move vertically along a fixed path with minimal friction force between the bar and the support rails. The Smith machine did not have any counterweight mechanism to mimic free-weight exercises [17]. Bar velocity was measured using a linear transducer (T-FORCE System 3.60, Ergotech Consulting SL, Murcia, Spain). This was carried out by providing auditory and visual feedback in real time at a sampling rate of 1000 Hz. The transducer automatically determined the eccentric and concentric phases of each repetition, and the propulsive phase was defined as that portion of the concentric phase during which the measured acceleration (a) is greater than acceleration due to gravity (i.e., a ≥ −9.81 m·s^−2^) [8].

The protocol proposed by Sanchez et al. [37] was used for the bench press. Participants laid on a flat bench in the supine position with their feet supported and their hands on the bar, separated slightly beyond shoulder width (5–7 cm). Grip width was measured so that it could be reproduced in subsequent series. The participants were instructed to lower the bar slowly (~0.70–0.50 m·s^−1^) [38] and in a controlled motion until reaching 1 cm from the upper part of the xiphoid process. They were then told to stop for approximately 1.5 s [35] until the evaluator instructed them to extend their arms at maximum velocity without raising their trunk or shoulders from the bench. This protocol was followed to avoid the rebound effect and provide more reproducible and consistent measurements.

The test started with a weight of 10 kg for women and 20 kg for men, and four repetitions were performed. The weight was progressively increased by 3 kg for women and 5 kg for men. Three repetitions were performed for each weight until the MPV attained was lower than 0.50 m·s^−1^ [8]. From then on, progressive increases of 1 kg for women and 2 kg for men were implemented, and only two repetitions were performed. This continued until the participants could only perform one repetition with an elbow extension of 180° and an MPV of less than or equal to 0.20 m·s^−1^. This value was considered the participant’s 1RM [39].

For the full squat, the subjects started in an upright position with their knees and hips extended. They kept their feet at the same distance as the width of their shoulders and the bar on the trapeziums at the level of the acromion [38]. This position was carefully verified so that it could be reproduced in subsequent series. Participants descended in a controlled manner at an average velocity of ~0.70–0.50 m·s^−1^ until reaching a fibula-femoral flexion angle of 35–40° along the sagittal plane [38] to achieve a full squat [40]. This was measured with a goniometer (Nexgen Ergonomics, Point Claire, QC, Canada). Once in this position, participants paused for 1.5 s, and on the evaluator’s instruction, an extension was performed at maximum velocity.

This exercise started with a weight of 20 kg for women and 30 kg for men. Four repetitions were performed. The weight was progressively increased by 5 kg for women and 10 kg for men until the MPV reached was less than 0.60 m·s^−1^. From that moment on, progressive increases of 3 kg for women and 5 kg for men were implemented, and only two repetitions were performed. This continued until the participants could only perform a single repetition with an extension of 180° and an MPV value of less than or equal to 0.20 m·s^−1^. This value was considered the 1RM [41].

Three minutes of rest was given between sets for loads lower than 80% of the estimated 1RM, and five minutes of rest was allowed between sets for loads greater than 80% of the estimated 1RM [35,41].

The MPV was determined seventy-two hours after the results of the 1RM test were obtained for the bench press and full squat. This was carried out for loads corresponding to 50%, 60%, 70% and 80% 1RM using a linear velocity transducer (T-FORCE System 3.60, Ergotech Consulting SL, Murcia, Spain). Two repetitions were performed at maximum velocity for each movement following the protocols described above. The rest period between different loading conditions was set to 3 min for lighter and medium loads, while 5 min was implemented between the heavier loads [42] which were carried out progressively. The bench press tests were run first. After 10 min of recovery, the full squat tests were performed.

### 2.3. Statistical Analysis

Descriptive analyses were performed for all variables. Additionally, coefficients of variation were calculated for the MPV variables at different 1RM percentages. Due to signs of normality, independent group comparisons between men and women were performed using Student’s *t*-test. Student’s *t*-test was also carried out for paired samples between the bench press and full squat MPVs among men and women. Finally, four regression models were carried out to estimate the chest and squat MPVs in men and women based on the weight lifted at different percentages of 1RM. Analyses were performed using SPSS version 25.0 (SPSS Inc., IBM Company, Armonk, NY, USA). The significance level was set at 5%.

## 3. Results

The general descriptive characteristics of the maximum strength and MPV of all participants are presented in Table 1. For all %1RM values and both movements, men presented significantly higher MPV values than women (*p* < 0.05).

When the association between %1RM and MPV in the bench press and full squat exercises was analyzed, a strong correlation was observed for the bench press (r = 0.872) and squat (r = 0.806) among women (Figure 1). Similar correlations were observed in the group of men: r = 0.880 for bench press, and r = 0.832 for the full squat (Figure 2).

Despite these strong correlations, considerable variability in the MPV was detected for all %1RM values between subjects for both exercises. The minimum and maximum MPV values and the corresponding coefficients of variation for all %1RM values are presented in Table 2.

In both the male and female groups, the observed MPV for the bench press was higher than that for the full squat for all %1RM values (Table 3).

## 4. Discussion

The primary purposes of this study, which considered bench press and full squat exercises, were to determine [2] the load-velocity relationship, [3] the behavior of the mean propulsive velocity (MPV) at each percentage of one repetition maximum (1RM) and [4] the variability of the MPV at each %1RM between subjects and groups. The most notable finding of this study is that, despite the strong correlation between the MPV and %1RM, significant variability was observed among the MPV values for the full squat and bench press movements for all considered %1RM values (50%, 60%, 70% and 80%) among a group of young male and female recreational athletes.

The results of the present study are consistent with those observed in several previous studies in highly trained athletes that identified strong correlations (R^2^ = 0.91–0.98) between load (in terms of %1RM) and MPV [7,8,10,25,38,43]. Therefore, the results of the present study confirm the available evidence that movement velocity can be used to accurately estimate the relative load (%1RM) regardless of an athlete’s gender, strength level and degree of training.

The results of the present study confirm the strong load-velocity correlation observed in several previous studies in highly trained athletes (R^2^ = 0.91–0.98) [7,8,10,25,38,43]. 

However, the reported MPV values of these investigations are higher than those of the present study at the different %1RM values, both in the bench press and in the full squat. This is the case both for the deep full squat [7,10,16,25,38] and bench press movements [8,17,43]. For example, the MPV values obtained in the present study for the deep squat at 50% 1RM correspond approximately to those obtained at 70% in the studies by Sanchez-Medina et al. [38] and Gonzalez-Badillo et al. [7], and at 60% in the research of Conceicao et al. [10]. For the bench press movement, the differences between the MPV values are maintained but relatively small (16% maximum).

On the other hand, although these studies were carried out with male athletes experienced in this type of exercise, noticeable differences were also observed among them in terms of the MPV. Such differences could be attributed to long-term training adaptations, the type of sport and differences in the composition of muscle fibers, as suggested by Izquierdo et al. [44].

Large differences between men and women were observed regarding their load-velocity profiles. In the present study, the MPV profiles of men and women were different for each %1RM. This aligns with the research of Balsamore et al. [25], who observed that the velocity associated with each %1RM during the military press exercise was greater for men than women. Similar differences were also observed in research comparing the velocity values for each %1RM between young adults (21.0 ± 1.6 years old) and adults (42.6 ± 6.7 years old) who performed strength training [27].

These findings indicate that an individual’s load-velocity profile may be significantly different from the average profile obtained in a group, supporting the use of individual load-velocity assessment to prescribe %1RM in the bench press and in the full squat in both men and women. In other words, the same MPV value can represent a different %1RM for each participant [11,25,27].

Several studies [29,30,45] indicate that, due to the limitations of the general load-velocity ratio equations, the individual load-velocity ratio is recommended to provide more accurate prescriptions of relative load (%1RM). These statements are confirmed by our study based on the interindividual differences in the MPV for the two movements, the significantly different %1RM values and the high coefficients of variation. Furthermore, the velocity associated with each %1RM is subject-specific, while the differences between subjects are accentuated at lower relative loads [10,25,38,43].

The results of the present study confirm the findings of Torrejon and Balsamore [42], indicating the need for individual evaluations of the load-velocity relationship instead of using generalized group equations. Additionally, our results are in line with the limitations reported by Sanchez et al. [38]. The researchers indicated that caution should be taken when using the equations provided here to calculate load (%1RM) from velocity (or vice versa) in other populations, such as female athletes, athletes with significantly different relative strength ratio (RSR) values, untrained persons, older adults and persons suffering from illness or disabilities. Torrejon and Balsamore [42] argued against the use of generalized group equations proposed with the objective of predicting %1RM based on the velocity recorded from a single load condition [7,8,10,16].

Some studies [44,46,47] have established that the maximum force is lower in the upper limbs than in the lower limbs, especially considering peak torque. However, there is little evidence suggesting that the same is true for velocity or MPV. In the present study, differences in the MPV between the upper and lower extremities were observed in men at all levels of %1RM; in women, such differences occurred only at 70–80% 1RM. This gender difference could be associated with the type of physical activity, sport or recreational practice performed, the size and length of the muscle, the architecture of the muscle fiber, the angle and physical properties of the myotendinous junction, the type of fiber or the number of parallel cross bridges [48].

The results of the present study are also supported by the classification of the participants according to their RSR. Weak differences were observed in the %1RM velocity between the strong and weak men (ES = 0.18), while moderate differences were detected between the strong and weak women (ES = 0.78). Moreover, small differences were observed in the slope of the loading velocity profiles between the strong and weak women (ES = −0.39) [42].

To the best of our knowledge, this is the first study to explore the load-velocity ratio associated with each %1RM of full squat and bench press movements among young male and female recreational athletes. Interestingly, the MPV values were higher for the lower limbs than the upper limbs among men but not among women.

The results of this study contribute to raising awareness of the specificity of the velocity rate associated with each %1RM in each subject, as the load-velocity relationship appears to be specific to each exercise. Physical trainers should evaluate the load-velocity relationship for each athlete instead of using general equations when designing and monitoring strength training programs for recreational athletes.

## 5. Conclusions

In conclusion, the results of this study provide additional support that the use of linear regression models is not recommended for the reliable and precise determination of the load-velocity profile in a population of young athletes who begin strength work. The determination of the load-velocity profile must be carried out directly and individually for a more accurate prescription of the %1RM. The differences in the load-velocity profile between participants of the same sex are important and increase when men are compared with women.

## Figures and Tables

**Figure 1 ijerph-19-06756-f001:**
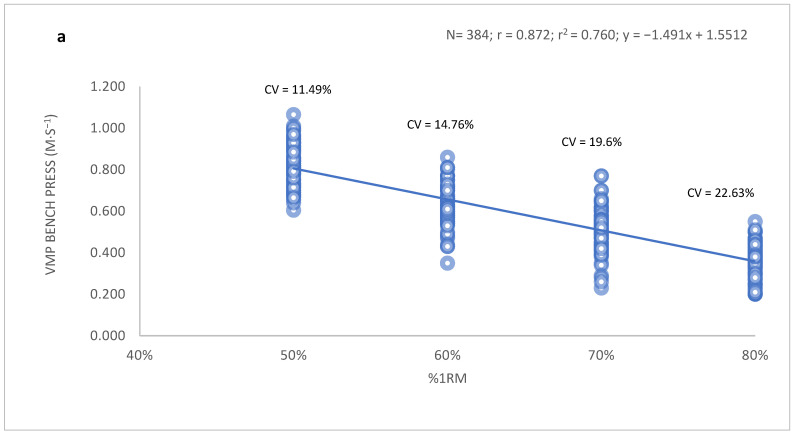
Relationship between MPV and relative load (%1RM) in the group of women for (**a**) the bench press and (**b**) full squat movements. Data were obtained from the values of the mobilization velocity of the isoinertial progressive load carried out in the sample of 96 young women.

**Figure 2 ijerph-19-06756-f002:**
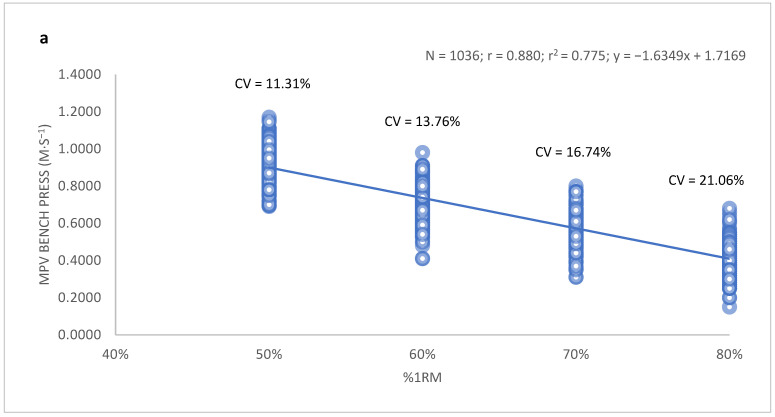
Relationship between MPV and relative load (%1RM) in the group of men for the (**a**) bench press and (**b**) full squat movements. Data were obtained from the values of the mobilization velocity of the isoinertial progressive load carried out in the sample of 259 young men.

**Table 1 ijerph-19-06756-t001:** Maximum strength and MPV values (mean and SD) in the full squat and bench press of the 352 participants.

Variables	Women (*n* = 96)	Men (*n* = 259)	*p*-Value
Maximum full squat strength (kg)	39.4 ± 10.8	67.5 ± 12.6	***
Maximum bench press strength (kg)	33.1 ± 8.0	59.3 ± 11.4	***
MPVSQ 50% 1RM (m·s^−1^)	0.81 ± 0.09	0.87 ± 0.08	*
MPVSQ 60% 1RM (m·s^−1^)	0.64 ± 0.11	0.70 ± 0.07	*
MPVSQ 70% 1RM (m·s^−1^)	0.55 ± 0.10	0.61 ± 0.09	**
MPVSQ 80% 1RM (m·s^−1^)	0.42 ± 0.11	0.49 ± 0.12	**
MPVSQ 100% 1RM (m·s^−1^)	0.23 ± 0.06	0.26 ± 0.11	*
MPVBP 50% 1RM (m·s^−1^)	0.82 ± 0.09	0.92 ± 0.10	*
MPVBP 60% 1RM (m·s^−1^)	0.63 ± 0.09	0.70 ± 0.10	*
MPVBP 70% 1RM (m·s^−1^)	0.51 ± 0.10	0.57 ± 0.10	*
MPVBP 80% 1RM (m·s^−1^)	0.36 ± 0.08	0.42 ± 0.09	*
MPVBP 100% 1RM (m·s^−1^)	0.19 ± 0.07	0.21 ± 0.10	*

Note. MPVSQ = mean propulsive velocity full squat; MPVBP = mean propulsive velocity bench press. *** *p* < 0.001, ** *p* < 0.01, * *p* < 0.05.

**Table 2 ijerph-19-06756-t002:** Maximum and minimum values and coefficient of variation of the MPV at the different percentages of 1RM in the bench press and full squat.

Variable	Women			Men		
	Minimum	Maximum	CV	Minimum	Maximum	CV
MPVPB50 (m·s^−1^)	0.61	1.07	11.49%	0.69	1.17	11.31%
MPVPB60 (m·s^−1^)	0.35	0.86	14.76%	0.41	0.98	13.76%
MPVPB70 (m·s^−1^)	0.23	0.77	19.60%	0.31	0.80	16.74%
MPVPB80 (m·s^−1^)	0.20	0.55	22.63%	0.15	0.68	21.06%
MPVSQ50 (m·s^−1^)	0.51	0.96	11.58%	0.59	1.05	9.26%
MPVSQ60 (m·s^−1^)	0.33	0.88	17.17%	0.46	0.94	11.20%
MPVSQ70 (m·s^−1^)	0.22	0.77	17.39%	0.42	0.90	14.10%
MPVSQ80 (m·s^−1^)	0.20	0.59	25.15%	0.20	0.88	24.20%

Note. MPVSQ = mean propulsive velocity full squat; MPVBP = mean propulsive velocity bench press.

**Table 3 ijerph-19-06756-t003:** Comparison of the MPV at the different percentages of 1RM in the bench press and full squat movements in men and women.

%1RM	Men			Women		
	MPVBP (m·s^−1^)	MPVSQ (m·s^−1^)	*p*-Value	MPVBP (m·s^−1^)	MPVSQ (m·s^−1^)	*p*-Value
50%	0.92 ± 0.10	0.87 ± 0.08	<0.001	0.82 ± 0.09	0.81 ± 0.09	NS
60%	0.70 ± 0.10	0.71 ± 0.08	NS	0.63 ± 0.09	0.64 ± 0.11	NS
70%	0.57 ± 0.09	0.61 ± 0.09	<0.001	0.51 ± 0.10	0.55 ± 0.10	***
80%	0.42 ± 0.09	0.49 ± 0.12	<0.001	0.36 ± 0.08	0.42 ± 0.11	***

Note. MPVSQ = mean propulsive velocity full squat; MPVBP = mean propulsive velocity chest press. *** *p* < 0.001.

## Data Availability

The data that supports the results of this information is found in the databases of the exercise physiology laboratory and is not publicly available due to the data protection law.

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
