# Peer review of "Bar Load-Velocity Profile of Full Squat and Bench Press Exercises in Young Recreational Athletes"

_ijerph, 2022, doi:10.3390/ijerph19116756_

Round 1
Reviewer 1 Report
Very good job on the study. I feel it was very well written and the science was solid. I have a couple of questions.
Did you do a power analysis?
Why was the Smith machine used?
Reviewer 2 Report
Thank you for the opportunity to review this manuscript which considers some interesting applied issues.
I would like to congratulate the authors for the work carried out and I present my concerns in order to improve the quality of the manuscript.
INTRODUCTION
Line 55 – As you remark, age and gender are important variables. One study comparing both variables found differences, highlighting the importance of individualization.
Alonso-Aubin, D. A., Chulvi-Medrano, I., Cortell-Tormo, J. M., Picón-Martínez, M., Rial Rebullido, T., & Faigenbaum, A. D. (2021). Squat and Bench Press Force-Velocity Profiling in Male and Female Adolescent Rugby Players. Journal of strength and conditioning research, 35(Suppl 1), S44–S50. https://doi.org/10.1519/JSC.0000000000003336
Line 59 – Maybe you should reference this sentence.
Banyard, H. G., Nosaka, K., Vernon, A. D., & Haff, G. G. (2018). The Reliability of Individualized Load-Velocity Profiles. International journal of sports physiology and performance, 13(6), 763–769. https://doi.org/10.1123/ijspp.2017-061
MATERIAL AND METHODS
Line 73 – Is VMP or MPV?
Line 74 – Why you do not recorded VMP at 90 and 100% 1-RM?
Line 98 – Did the bar reach the full elbow extensión in the concentric phase? In the final position, back of the athlete was on the bench?
Line 103 – V is velocity?
Line 112 – Which method? Please, reference.
Line 122 – Change Schmitt machine to Smith machine.
Line 155 – 0.20m/s-1
Why do you consider 0.20m/s-1 as the 1-RM? Do you think that assuming this data as valid for the entire sample could compromise the validity of the results?
In the introduction you highlight the importance of the possible differences between subjects based on variables such as age or sex and 0.20m/s-1 as the 1-RM could be different for each subject.
RESULTS
Line 177 to 179 and 76-78 – I think you should delete as the data is presented previously.
Table 1 – You should highlight the significant differences with *.
Table 2 – I suggest to add the data in table 1.
Table 3 - You should highlight the significant differences with *.
DISCUSSION
The discussion presents a good common thread and responds to the objectives of the study.
Line 243- You could check this paper for speed profiles in males and females.
Alonso-Aubin, D. A., Picón-Martínez, M., & Chulvi-Medrano, I. (2021). Strength and Power Characteristics in National Amateur Rugby Players. International journal of environmental research and public health, 18(11), 5615. https://doi.org/10.3390/ijerph18115615
Author Response
Response to Reviewer 2 Comments
We would like to thank you for your comments, suggestions and contributions which, when reflecting on them, significantly improved the quality of the manuscript.
Point 1: Line 55 – As you remark, age and gender are important variables. One study comparing both variables found differences, highlighting the importance of individualization.
Response 1:
Gender and age are important variables in this context. There are few studies involving men and women, in which the speed associated with each % of 1RM was greater in men (17), and significant differences were identified for maximum squat speed at 40, 50, 60 and 70 % of 1RM; 1RM bench press max power at 50%, 60%, 70%, and 80%; the time to reach the maximum speed of 1RM in the bench press at 50, 60, 70 and 80%; and time to reach 1RM bench press max power at 50%, 60%, 70%, and 80%. (18)
Point 2: Line 59 – Maybe you should reference this sentence.
Response 2: Therefore, it is important to accurately determine individualized load-velocity profiles (19)
Point 3: Line 73 – Is VMP or MPV?
Response 3: The change was made, it is MPV
Point 4: Line 74 – Why you do not recorded VMP at 90 and 100% 1-RM?
Response 4: The tests at 90% and 100% 1RM were not performed because the study participants had no experience with strength training and several of the studies also perform it up to 80% because these are the ranges where the subjects train. The values of the VMP at 100% were added to the table when the maximum force was determined.
Point 5: Line 98 – Did the bar reach the full elbow extensión in the concentric phase? In the final position, back of the athlete was on the bench?
Response 5: In the bench press, special emphasis was placed on the controlled descent of the bar until 90° elbow flexion, the width of the bar grip was the biacromial distance, and in this position they had to wait (approximately 1.5 seconds) until the evaluator's order to perform the movement that had to be executed at the highest possible speed keeping the back on the bench and until reaching full extension.
Point 6: Line 103 – V is velocity?
Response 6: The V was eliminated because the speed analysis was not performed, only the VMP
Point 7: Line 112 – Which method? Please, reference.
Response 7: Standard procedures for testing 1RM in youth were followed (25). . Faigenbaum AD, Milliken LA, Westcott WL. Maximal strength testing in healthy children. J Strength Cond Res 17: 162–166, 2003.
Point 8: Line 122 – Change Schmitt machine to Smith machine.
Response 8: It has been made
Point 9: Line 155 – 0.20m/s-1
Response 9: : It is expressed this way because most articles express it that way m.s -1
Point 10: Why do you consider 0.20m/s-1 as the 1-RM? Do you think that assuming this data as valid for the entire sample could compromise the validity of the results
Response 10: In previous studies, both ours and those of other authors, it has been identified that when a load cannot be mobilized beyond 0.20 m.s -1 it is considered one of the criteria to determine the 1RM. However, this data was only used as a guide to determine the 1RM of the participants, but it was not a value that everyone had to meet, as you rightly mention, it would compromise the validity of the results. The average value in the VMP obtained was 0.19 in women and 0.21 in men.
Point 11: Line 177 to 179 and 76-78 – I think you should delete as the data is presented previously.
Response 11: They were removed
Point 12: Table 1 – You should highlight the significant differences with *.
Response 12: It has been made
Point 13: Table 2 – I suggest to add the data in table 1.
Response 13: It has been made
Point 14: Table 3 - You should highlight the significant differences with *.
Response 14: It has been made
Point 15: Line 243- You could check this paper for speed profiles in males and females.
Response 15: These findings indicate that the upload speed profiles of an individual can be significantly different from the average profile obtained from a group. This confirms the existence of individual upload speed profiles; in other words, the same MPV absolute value may represent a different %1RM for each participant, as observed in different previous studies (17, 18, 33)

Reviewer 3 Report
This paper is about the load velocity relationship during bench press and squat. 69 female and 256 men were performing incremental test. This is an impressive number of participants. There was only a low linear behaviour of the load velocity relationship. Therefore the authors concluded that an individual assessment is requirement for recreational athletes.
Intro.
l32. Please include a reference.
l37. There is much more literature available. Please expand this section and also include the difference of the mathematical models. Here the linearity but also different approaches should be presented.
l38. This sentence should be deleted, it does not add to the introduction.
l40-41. Many different researchers have performed studies in this area. It would be more valuable if these reference would be included.
l56. there are different studies available including men and women. They should be included here.
General Comment. Sometime the word „Speed“ is used, sometime „velocity“. If you make a difference in the context of speed/velocity -load relationship, this should be addressed in the introduction. If not please use the same word.
Method.
How were the participants instructed to move with maximal velocity? was there a verbal motivation too?
Please introduce MPV, V and peak power properly. Why were these values chosen? In athletes sometimes the average velocity respectively the peak velocity produces the better 1RM estimate.
Was a Bonferroni correction performed for the multiple t.tests?
Results.
l 177.179 this should be included in the method section. please round your values up to meaningful.
Table 1. were the values normalized to the body weight? This probably would make sense.
Fig 1. please introduce pecho / sentadilla . probably it would make sense to use english words instead?
Would it make sense to normalize the velocity with the body / segment length?
Fig 2 only men, why is n=1036? the values at .5, .6, .7, .8 are belonging to the same person.
Would it be possible instead to have a figure with men and women in different colors? and 3 line for the regression, one both, one m and one f?
Table 3. is the p value for the comparison m-f or BP-SQ? It probably should be for m vs. f
Author Response
Response to Reviewer 3 Comments
We would like to thank you for your comments, suggestions and contributions which, when reflecting on them, significantly improved the quality of the manuscript.
Point 1: l32. Please include a reference.
Response 1: It was included
Point 2: l37. There is much more literature available. Please expand this section and also include the difference of the mathematical models. Here the linearity but also different approaches should be presented.
Response 2: The general load-velocity relationship was introduced by Gonzalez-Badillo and Sánchez-Medina (6) who used a second-order polynomial regression equation to estimate the %1RM based on velocity during the bench press exercise.
The mathematical models use linear regressions between the known values of load (X) and average velocity (AV) (Y) and from this linear regression, the slope (SLP), the theoretical AV at 0 kg (AVO) and the theoretical load at speed 0 m/s-1 (LD0). The arguments for the use of these variables is that VA better represents the subject's ability to move the load throughout the concentric phase and that VA decreases linearly with increasing load, which facilitates mathematical analysis. (5)
Point 3 : l38. This sentence should be deleted, it does not add to the introduction
Response 3: The change was made
Point 4: l40-41. Many different researchers have performed studies in this area. It would be more valuable if these reference would be included.
Response 4: The change was made
Point 5: l56. there are different studies available including men and women. They should be included here
Response 5: The change was made
Point 6: General Comment. Sometime the word „Speed“ is used, sometime „velocity“. If you make a difference in the context of speed/velocity -load relationship, this should be addressed in the introduction. If not please use the same Word.
Response 6: The change was made and left in all velocity and the load-velocity relationship
Point 7: How were the participants instructed to move with maximal velocity? was there a verbal motivation too?
Response 7: During the familiarization phase, the subjects were instructed to perform concentric actions at maximum speed and during the tests, the participants were verbally motivated to move the load at maximum speed in each repetition.
Point 8: Please introduce MPV, V and peak power properly. Why were these values chosen? In athletes sometimes the average velocity respectively the peak velocity produces the better 1RM estimate.
Response 8: Recently, the mean propulsive velocity (MPV) has been used to estimate the relative intensity of the strength training and establish the velocity attained with the 1RM load (badillo 6, loturco , Jidovtseff,) and in the study by Conceicao that compared the estimation of %1RM from speed and MVP, they observed: “For all exercises, a strong relationship between Vmax and the %1RM was found: leg press (r 2 adj = 0.96; 95 % CI for slope is [−0.0244, −0.0258], P < 0.0001), full squat (r 2 adj = 0.94; 95% CI for slope is [−0.0144, −0.0139], P < 0.0001) and half squat (r 2 adj = 0.97, 95% CI for slope is [−0.0135, −0.00143], P< 0.0001), as shown in Figure 3. Similar results were observed in the MPV and the %1RM relationship: leg press (r 2 adj = 0.96, 95% CI for slope is [−0.0169, −0.0175], P < 0.0001), full squat (r 2 adj = 0.95, 95% CI for slope is [−0.0136, −0.0128], P < 0.0001) and half squat (r 2 adj = 0.96; 95% CI for slope is [−0.0116, −0.0124], P < 0.0001). Therefore the use of either of the two allows us a similar estimate of 1RM
Point 9: Was a Bonferroni correction performed for the multiple t.tests?
Response 9: Bonferroni posthoc tests are not carried out, considering that the only comparisons are between men and women in the variables of maximum repetition, percentage of RM and bench press with squat. This type of test is not necessary, nor is it necessary to penalize the value of significance. These types of corrections are applied when more than two groups are compared.
Point 10: 177.179 this should be included in the method section. please round your values up to meaningful.
Response 10: The change was made
Point 11: Table 1. were the values normalized to the body weight? This probably would make sense.
Response 11: To observe the normality of the variables, they were normalized with the values of body weight and a normal distribution was identified.
Point 12: Fig 1. please introduce pecho / sentadilla . probably it would make sense to use english words instead?
Response 12: The change was made
Point 13: Would it make sense to normalize the velocity with the body / segment length?
Response 13: It has been identified that anthropometric dimensions can have an effect on MPV. (Study by Fahs, C. A., Blumkaitis, J. C., & Rossow, L. M. (2019). Factors related to average concentric velocity of four barbell exercises at various loads. The Journal of Strength & Conditioning Research, 33(3), 597-605. The present study did not have that purpose, therefore body segments were not measured.
Point 14: Fig 2 only men, why is n=1036? the values at .5, .6, .7, .8 are belonging to the same person.
Response 14: The values 5,6,7,8 correspond to the %1RM 50%,60%,70%,80% and in each of these percentages all 259 men are found. The value of n=1036 corresponds to the total of measurements made. That is, 259 men who performed the tests in 4 different RM percentages, 50, 60, 70 and 80. Therefore 259*4=1036. This type of analysis has been used in other studies: González-Badillo, J. J., & Sánchez-Medina, L. (2010). Movement velocity as a measure of loading intensity in resistance training. International journal of sports medicine, 31(05), 347-352.
Point 15: Would it be possible instead to have a figure with men and women in different colors? and 3 line for the regression, one both, one m and one f?
Response 15: On three of the four charts the trend lines almost overlap each other, so it is better to show them separately.
Point 16: Table 3. is the p value for the comparison m-f or BP-SQ? It probably should be for m vs. f
Response 16: Table 3 presents the differences between the VMP in the SQ and the BP in the different % of 1RM in both men and women.

Reviewer 4 Report
This study investigated the load-velocity profile of the squat and bench press exercise in a population of untrained men and women. The authors compared the mean propulsive velocity at various loads between exercises and between sexes. The authors should be commended for including such a robust sample. However, this paper needs significant editing from a native English speaker. I also have concerns regarding novelty the main conclusion – that load-velocity profiles need to be individualized. This is already well established and is not a novel finding. The authors state that this is the first study that investigated the load-velocity profile in men and women but this has been examined in studies that are not referenced in the current manuscript (specific comments below). Another concern I have is why specifically is studying the load-velocity profile in untrained individuals needed? The authors did provide adequate familiarization with the movements in my opinion but why would a novice need to use velocity to determine training loads when they should be rapidly progressing during the initial training period regardless? Finally, the use of %CV is misleading when looking at variability between loads since the mean velocity decreases with higher loads. All things considered, I am not sure this manuscript can be improved to meet the standard for a peer-reviewed publication in a journal but could be a data set presented at a conference. Below are my specific questions and concerns.
- Abstract, purpose – I am not sure what is meant by ‘behavior’ of the mean propulsive velocity, Could this simply be stated as, “to determine the mean propulsive velocity (MPV) in each percentage of a one-repetition maximum (1RM)”?
- Full squat and complete squat are used interchangeably, please change to one for consistency.
- Speed and velocity are used interchangeably, please change to one for consistency.
- Abstract, results – the variability is identified as high (CV17%) but it is unclear to what this value refers – between groups, within groups, at what %RM?
- Introduction, line 52 – “The equal absolute speed represents different intensities or % 1RM values depending on the exercise performed ”. Can the authors clarify what is meant by “the equal absolute speed”?
- Introduction, line 58 – “Finally, an individual’s load-velocity profile depends on their training level and strength-working experience.” – Can the authors provide a reference for this statement? A relevant study that has previously looked at factors related to the load-velocity relationship is:
- Fahs, C. A., Blumkaitis, J. C., & Rossow, L. M. (2019). Factors related to average concentric velocity of four barbell exercises at various loads. The Journal of Strength & Conditioning Research, 33(3), 597-605.
- Introduction, line 63 – “To the best of our knowledge, no previous study has determined the load-velocity relationship between bench press and full squat exercises in a group of recreational male and female athletes ” – The study referenced above by Fahs et al. (2019) did examine the load-velocity relationship in men and women for the squat and bench press exercise. Sex differences in the load-velocity profile of some lifts have also been identified:
605. Kasovic, J., Martin, B., & Fahs, C. A. (2019). Kinematic differences between the front and back squat and conventional and sumo deadlift. The Journal of Strength & Conditioning Research, 33(12), 3213-3219.
606. Torrejón, A., Balsalobre-Fernández, C., Haff, G. G., & García-Ramos, A. (2019). The load-velocity profile differs more between men and women than between individuals with different strength levels. Sports Biomechanics, 18(3), 245-255.
- Introduction, line 65 – “Therefore, detailed studies are required to identify the behavior of the load-velocity relationship in an untrained population” Can the authors clarify why would untrained individuals need to know about their load-velocity profile when the application for velocity-based training is generally for athletes and well-trained lifters?
- VMP and MPV are used interchangeably, please change for consistency.
- Methods, line 96 – “the bar descended 1 cm from the upper part during the xiphoid process.” Please clarify this sentence.
- Methods – “maximal force” is mentioned in relation to the assessment of the 1RM. This is not accurate as the 1RM is limited by the weakest point in the range of motion. Please revise.
- Methods – Schmitt and Smith machines are used interchangeably. Please revise.
- Methods, line 115 – the linear transducer was used to control or measure velocity? Please clarify.
- Methods, line 116 – what is “displacement velocity”? Displacement and velocity are two different variables, please clarify.
- Methods, line 119 – “as well as the driving phase of the concentric phase during which the acceleration of the bar is greater than the acceleration due to gravity”. I assume the acceleration of the bar during the entire repetition would be greater than the acceleration due to gravity as the lifter is applying an upward force to the bar at all times during the lift. Please revise.
- Methods, line 137- “only perform one repetition with an extension of 180°” – does this refer to the extension of the elbow to 180 degrees? Please clarify.
- Methods, line 159 – it is stated that the velocity, MPV, and peak power were all obtained from the tests. Why is only MPV data presented?
- Methods, line 162-163 – were the two repetitions at each load (50-60-70-80% 1RM) performed consecutively or with some rest interval in between each rep? What rest interval was allotted between each load? Were the loads implemented in a progressive order?
- Methods, line 167 – why is the coefficient of variation used to compare variability across loads? As the mean velocity declines with load, CV will increase even if the standard deviation remains the same. Please clarify.
- Methods, line 171 – “chest” should be “bench press”.
- Results – participant characteristics are already detailed in the methods.
- Results, line 181 – “higher MPV values than women, but these differences are not significant ”. According to Table 1, all p-values are less than or equal to 0.05. Please do not round p-values to less than 3 decimal places and clarify if these p-values do or do not exceed your alpha level.
- Figures 1 and 2 – please translate the figure's axis labels into English. There are an unnecessarily high number of decimal places in the values listed on the y-axis. The x-axis should be labeled as 40, 50, 60, 70, 80 rather than 0.4, 0.5, etc. as the units are %1RM. Also, why not include the %CV at each load in Figure 1?
- Discussion lines 219-220 – “Although this relationship is indeed ratified, the MPV values in these investigations are higher than in the present study for all % 1RM ”. This sentence needs to be clarified in starting this paragraph. What relationship is being described and what other investigations are being referenced?
- Discussion, general – the discussion is difficult to follow. Each paragraph needs a clear topic sentence to begin followed by an explanation in the subsequent sentences.
- Discussion, line 233 – “with men exhibiting more pronounced load-velocity profiles than women ”. What does a “more pronounced load-velocity profile” mean?
- Discussion, line 240 – “individual’s charge speed profiles ” – please clarify this.
- Discussion, line 257 – please define “RSR”.
- Discussion, line 271 – please define “TRF”.
Author Response
Response to Reviewer 4 Comments
We would like to thank you for your comments, suggestions and contributions which, when reflecting on them, significantly improved the quality of the manuscript.
Point 1: Abstract, purpose – I am not sure what is meant by ‘behavior’ of the mean propulsive velocity, Could this simply be stated as, “to determine the mean propulsive velocity (MPV) in each percentage of a one-repetition maximum (1RM)”?
Response 1 : The purpose of this study was to determine the mean propulsive velocity (MVP) at various percentages of one repetition maximum (1RM) in the full squat and chest press exercises.
Point 2 : Full squat and complete squat are used interchangeably, please change to one for consistency.
Response 2: The adjustment was made and the entire document was left full squat.
Point 3 : Speed and velocity are used interchangeably, please change to one for consistency.
Response 3: The adjustment was made and velocity remained throughout the document.
Point 4 : Abstract, results – the variability is identified as high (CV17%) but it is unclear to what this value refers – between groups, within groups, at what %RM?
Response 4: High variability in MPV (11.49% to 22.63) in bench press and full squat (11.58% to 25.15%) was observed in women and men (11.31% to 21.06). % and 9.26% to 24.2% ) in the different percentages of 1RM evaluated.
Point 5 : Introduction, line 52 – “The equal absolute speed represents different intensities or % 1RM values depending on the exercise performed ”. Can the authors clarify what is meant by “the equal absolute speed”?
Response 5: It was modified due to an error in the wording. The same speed represents different intensities or %1RM values depending on the exercise performed.
Point 6 : Introduction, line 58 – “Finally, an individual’s load-velocity profile depends on their training level and strength-working experience.” – Can the authors provide a reference for this statement?
Response 6: A relevant study that has previously looked at factors related to the load-velocity relationship is: 605. Fahs, C. A., Blumkaitis, J. C., & Rossow, L. M. (2019). Factors related to average concentric velocity of four barbell exercises at various loads. The Journal of Strength & Conditioning Research, 33(3), 597-605.
Point 7 : Introduction, line 63 – “To the best of our knowledge, no previous study has determined the load-velocity relationship between bench press and full squat exercises in a group of recreational male and female athletes ”
Response 7: In the studies reviewed, the participants had experience in strength training. In our case, the study participants were recreational athletes who had no previous experience in strength training. Precisely the study indicates that the use of the load-velocity values established in the population of subjects trained in strength is not applicable to the population of recreational athletes or young people who start a strength training program.
The study by Fahs et al. (2019) participants had an average of one year of strength training experience, similar to the study by Kasovic, J., Martin, B. and Fahs, C. A. (2019). It was also carried out with subjects with experience in strength training.
Also in the study by Torrejón, A., Balsalobre-Fernández, C., Haff, G. G., & García-Ramos, A. The male participants presented an experience with the bench press exercise (6.2 ± 2.0 and women 1.2 ± 1.5 years.). The most recent study by Pérez-Castilla, García-Ramos, Padial, Morales-Artacho and Feriche J (Load-Velocity Relationship in Variations of the Half-Squat Exercise: Influence of Execution Technique. Strength Cond Res 2020 Apr;34(4) :1024-1031.doi: 10.1519/JSC.0000000000002072.) the participants had 3.0 ± 1.6 years of resistance training experience.
Point 8 : Introduction, line 65 – “Therefore, detailed studies are required to identify the behavior of the load-velocity relationship in an untrained population” Can the authors clarify why would untrained individuals need to know about their load-velocity profile when the application for velocity-based training is generally for athletes and well-trained lifters?
Response 8: In modern sport, strength is a fundamental component of training, regardless of the level at which the sport is practiced. A young athlete beginning his performance career requires knowledge of his load-velocity profile for his strength training program. On the other hand, strength training goes beyond the context of sport and is involved in the context of health. It is sufficiently documented that strength is associated with risk factors for death from multiple diseases. Therefore, in the strength training population from a health perspective it is equally important to know your velocity load profile for your strength training program.
Point 9 : VMP and MPV are used interchangeably, please change for consistency.
Response 9: The change was made
Point 10: Methods, line 96 – “the bar descended 1 cm from the upper part during the xiphoid process.” Please clarify this sentence.
Response 10: The change was made
Point 11: Methods – “maximal force” is mentioned in relation to the assessment of the 1RM. This is not accurate as the 1RM is limited by the weakest point in the range of motion. Please revise.
Response 11: The change was made
Point 12: Methods – Schmitt and Smith machines are used interchangeably. Please revise.
Response 12: The change was made
Point 13: Methods, line 115 – the linear transducer was used to control or measure velocity? Please clarify.
Response 13: The change was made. Bar velocity was measured using a linear transducer
Point 14 : Methods, line 116 – what is “displacement velocity”? Displacement and velocity are two different variables, please clarify.
Response 14: The change was made
Point 15 : Methods, line 119 – “as well as the driving phase of the concentric phase during which the acceleration of the bar is greater than the acceleration due to gravity”. I assume the acceleration of the bar during the entire repetition would be greater than the acceleration due to gravity as the lifter is applying an upward force to the bar at all times during the lift. Please revise.
Response 15: The propulsive phase was defined as that portion of the concentric phase during which the measured acceleration ( a ) is greater than acceleration due to gravity (i. e. a ≥ − 9.81 m · s − 2 ). Badillo
The propulsive phase was defined as that part of the concentric phase during which the measured acceleration ( a ) is greater than the acceleration due to gravity (ie, a ≥ − 9.81 m s − 2 ).
Point 16: Methods, line 137- “only perform one repetition with an extension of 180°” – does this refer to the extension of the elbow to 180 degrees? Please clarify.
Response 16: Adjusted, elbow extension at 180 degrees
Point 17: Methods, line 159 – it is stated that the velocity, MPV, and peak power were all obtained from the tests. Why is only MPV data presented?
Response 17: As the objective for the present study was to determine the load-velocity profile, maximum power data were not reported. For this reason it was removed from the text.
Point 18: Methods, line 162-163 – were the two repetitions at each load (50-60-70-80% 1RM) performed consecutively or with some rest interval in between each rep? What rest interval was allotted between each load? Were the loads implemented in a progressive order?
Response 18: The rest period between different loading conditions was set to 3 min for lighter and medium loads, while 5 min were implemented between the heavier loads (Torrejon) and were carried out progressively.
Point 19 : Methods, line 167 – why is the coefficient of variation used to compare variability across loads? As the mean velocity declines with load, CV will increase even if the standard deviation remains the same. Please clarify.
Response 19: The coefficient of variation is a measure of dispersion that allows comparing the dispersion of two samples regardless of their units (González, M. Á. M., Villegas, A. S., Atucha, E. T., & Fajardo, J. F. (Eds.). (2020) ). Friendly biostatistics. Elsevier.). Therefore, it is more appropriate to use the CV than the standard deviation in the comparison between loads.
Point 20 : Methods, line 171 – “chest” should be “bench press”.
Response 20: The change was made
Point 21: Results – participant characteristics are already detailed in the methods.
Response 21: Removed from paragraph
Point 22: Results, line 181 – “higher MPV values than women, but these differences are not significant ”. According to Table 1, all p-values are less than or equal to 0.05. Please do not round p-values to less than 3 decimal places and clarify if these p-values do or do not exceed your alpha level.
Response 22: The wording was changed
For all % 1RM and both movements, men presented significantly higher MPV values than women (p<0.05)
Point 23: Figures 1 and 2 – please translate the figure's axis labels into English. There are an unnecessarily high number of decimal places in the values listed on the y-axis. The x-axis should be labeled as 40, 50, 60, 70, 80 rather than 0.4, 0.5, etc. as the units are %1RM. Also, why not include the %CV at each load in Figure 1?
Response 23: The change was made
Point 24: Discussion lines 219-220 – “Although this relationship is indeed ratified, the MPV values in these investigations are higher than in the present study for all % 1RM ”. This sentence needs to be clarified in starting this paragraph. What relationship is being described and what other investigations are being referenced?
Response 24: The results of the present study confirm the strong load-velocity correlation observed in several previous studies in highly trained athletes (R2 = 0.91-0.98) (6, 7, 17, 25, 27, 31). However, the reported MPV values of these investigations are higher than those of the present study in the different % 1RM, both in bench press and in full squat.
Point 25: Discussion, general – the discussion is difficult to follow. Each paragraph needs a clear topic sentence to begin followed by an explanation in the subsequent sentences.
Response 25: The wording was changed
Point 26: Discussion, line 233 – “with men exhibiting more pronounced load-velocity profiles than women ”. What does a “more pronounced load-velocity profile” mean?
Response 26: The wording was changed
Point 27: Discussion, line 240 – “individual’s charge speed profiles ” – please clarify this.
Response 27: These findings indicate that an individual's load-velocity profiles may be significantly different from the average profile obtained in a group, supporting the use of individual load-velocity assessment to prescribe %1RM in the bench press. and in the full squat in both men and women. In other words, the same MPV value can represent a different 1RM% for each participant (17, 18,33)
Point 28: Discussion, line 257 – please define “RSR”.
Response 28: Definition: relative strength ratio
Point 29: Discussion, line 271 – please define “TRF”.
Response 29: The relative strength ratio RSR was adjusted, the initials in Spanish had been left.

Round 2
Reviewer 3 Report
Thank you very much for your revision. You have included the following statement:“The general load-velocity 39 relationship was introduced by Gonzalez-Badillo and Sanchez-Medina (7) who used a 40 second-order polynomial regression equation to estimate %1RM during the bench press 41 exercise based on velocity. *
I personally would not feel comfortable with this as an author. In google scholar there are more than 5000 hits with the key words: strength training, 1RM and velocity in the time between 1900 and 2010.
Please be aware that your work is not smaller when it reflects previous studies. It shows that you know the field well and what other groups are doing. This with strengthen your publications a lot.
Author Response
Point 1
You have included the following statement: “The general load-velocity 39 relationship was introduced by Gonzalez-Badillo and Sanchez-Medina (7) who used a 40 second-order polynomial regression equation to estimate %1RM during the bench press 41 exercise based on velocity. *
I personally would not feel comfortable with this as an author. In google scholar there are more than 5000 hits with the key words: strength training, 1RM and velocity in the time between 1900 and 2010.
Please be aware that your work is not smaller when it reflects previous studies. It shows that you know the field well and what other groups are doing. This with strengthen your publications a lot.
Response 1
Based on your valuable suggestion, the following references were included:
